# Cost-Effectiveness of Vaccination with the 20-Valent Pneumococcal Conjugate Vaccine in the Italian Adult Population

**DOI:** 10.3390/vaccines10122032

**Published:** 2022-11-28

**Authors:** Barbara Polistena, Giancarlo Icardi, Andrea Orsi, Federico Spandonaro, Roberto Di Virgilio, Daniela d’Angela

**Affiliations:** 1C.R.E.A. Sanità, Roma and University of Roma “Tor Vergata”, 00133 Rome, Italy; 2Department of Health Sciences (DISSAL), IRCCS Ospedale Policlinico San Martino, University of Genoa, 16132 Genova, Italy; 3C.R.E.A. Sanità, University San Raffaele, 00166 Rome, Italy; 4Pfizer Italia s.r.l., 00188 Rome, Italy

**Keywords:** vaccines, immunization, cost consequences, cost-effectiveness, cost-utilities

## Abstract

The availability of a new 20-valent pneumococcal conjugate vaccine (PCV) makes it appropriate to assess its cost-effectiveness. This was evaluated by adopting the Italian National Health Service perspective, using a cost consequences Markovian model. The expected effects of vaccination with 20-valent PCV were compared with the administration of 13-valent PCV and 15-valent PCV. Assuming a 100% vaccination of cohorts aged 65–74 years, in the (lifetime) comparison between 20-valent PCV and 13-valent PCV, the former is dominant (lower cost for a better health outcome). A reduction in disease events was estimated: −1208 deaths; −1171 cases of bacteraemia; −227 of meningitis; −9845 hospitalised all-cause nonbacteremic pneumonia cases (NBP) and −21,058 non-hospitalised. Overall, in the Italian population, a total gain of 6581.6 life years and of 4734.0 QALY was estimated. On the cost side, against an increase in vaccinations costs (EUR +40.568 million), other direct health costs are reduced by EUR 48.032 million, with a net saving of EUR +7.464 million. The comparison between 20-valent PCV and 15-valent PCV results in an Incremental Cost-Effectiveness Ratio (ICER) of EUR 66 per life year gained and EUR 91 per QALY gained. The sensitivity analyses confirm the robustness of the results. We can conclude that the switch to 20-valent PCV is a sustainable and efficient investment.

## 1. Introduction

*Streptococcus pneumoniae* or pneumococcus is a pathogen responsible for diseases of varying severity that are one of the major causes of mortality and morbidity worldwide. Pneumococcal infections affect people of all ages, but children under two years of age and adults 65 years of age and over are at the highest risk.

Based on the polysaccharide characteristics of the capsule with which it is coated, over 90 globally widespread serotypes are classified, of which about 40 are capable of generating infections in humans. However, only a limited number of them cause severe infections, called invasive pneumococcal infections, in various age groups of the world population. The distribution of invasive *S. pneumoniae* serotypes changes according to different geographical areas. In 2000, the World Health Organization (WHO) [1] detected 14.5 million cases of invasive infections globally. In the USA, in 2019, the Center for Disease Control (CDC) [2] estimated that about 150,000 hospitalisations were caused by pneumococcal disease. In Italy, in 2019 and 2020 (the latter is the year in which data are influenced by the effects of the COVID-19 pandemic on the healthcare services), through a surveillance system active since 2007, 1679 and 499 cases of invasive pneumococcal disease (IPD) were identified, respectively, with an incidence (number of new cases among the population) of 2.81/0.84 per 100,000 inhabitants, respectively. It should be considered that IPD cases are reported in the surveillance database by hospitals or regional health authorities on a voluntary basis [3]. Since the starting of the surveillance, the Italian regions have shown a different propensity to report cases and to submit the pneumococcal isolates to serotyping; therefore, a certain level of under-reporting of IPD cases cannot be excluded.

Today, in Europe and the USA, the incidence of invasive infections is about 100 cases per 100,000 inhabitants. Generally, in temperate climates the incidence of pneumococcal infections peaks in winter months [4,5,6,7,8].

Transmission of *S. pneumoniae* occurs from person to person through direct contact or by inhaling respiratory droplets produced by talking, coughing, and sneezing. It is estimated that 20 to 40% of children and 5 to 10% of adults are asymptomatic carriers of the bacterium in the nasopharynx [9], the only known reservoir in humans. The incubation period is uncertain, but is assumed to be about 1–3 days. It is most likely to spread in indoor social settings, such as nursing homes, long-term care facilities, hospital wards, prisons, military bases, universities or schools, homeless dormitories, and kindergartens [10].

Pneumococcal disease can include many different types of infections. Most pneumococcal infections are mild; however, some of them can be fatal or cause long-term problems [1]. Invasive pneumococcal disease is the term used for the most severe pneumococcal infections, particularly bacteraemia, sepsis, and meningitis.

The risk of developing a pneumococcal infection is higher in the following cases: chronic diseases (such as heart and lung disorders, diabetes, and liver diseases); alcohol-addiction; disorders that weaken the immune system, such as HIV infection; drugs that suppress the immune system, such as corticosteroids or chemotherapy drugs; absence of a functioning spleen; sickle cell anaemia; stay in a long-term care facility; and smoking [10].

Currently, the National Vaccine Prevention Plan (PNPV) strongly recommends pneumococcal vaccination in the first year of life (for those born from 2012 onwards), envisaging three doses and a coverage target greater than or equal to 95% in newborns. As for the adult population, vaccination is recommended in people 65 years of age, providing for two doses (a first dose of 13-valent conjugate vaccine and a second dose of 23-valent polysaccharide vaccine, at least 2 months apart) and a coverage target of at least 75% of the people aged 65 years.

Conjugate vaccines have ensured significant protection against pneumococcal infection with effects extending to all age groups through the induction of herd immunity. At the same time, the limited serotype coverage of conjugate vaccines led to a phenomenon of partial replacement, i.e., substitution of part of the circulating serotypes. Therefore, though the use of pneumococcal conjugate vaccines (PCVs) has greatly reduced PCV vaccine-type disease, there continues to be a clinical and economic burden of pneumococcal disease.

While vaccination coverage in the paediatric population has been stable around the coverage target for more than a decade, vaccination coverage data in adults are not systematically and uniformly collected. The available data are derived from local or regional studies, and show that coverage is suboptimal [11,12].

In 2022, based on the results of a large programme of Phase 1, 2, and 3 clinical trials, European Medicine Agency (EMA) also approved a 20-valent, single-dose polysaccharide pneumococcal conjugate vaccine administered intramuscularly, indicated for protecting adults aged 18 years and over against 20 serotypes responsible for most cases of invasive disease. It is an evolution of the 13-valent vaccine as it contains seven additional capsular polysaccharides associated with invasive pneumococcal disease with high mortality rates and antibiotic resistance. The results of Phase 3 studies in children are expected [13].

The availability of a new vaccine, and the knowledge gap on cost-effectiveness of its adoption, makes it appropriate to produce a first (to our knowledge) assessment of its cost-effectiveness in the Italian adult population, which is the primary goal of this analysis. 

## 2. Materials and Methods

The 20-valent pneumococcal conjugate vaccine (PCV20) cost-effectiveness in adults was evaluated by adopting the National Health Service (NHS) perspective and adapting a cost-effectiveness model developed for Pfizer Inc. by Policy Analysis Inc. (Brookline, MA, USA) to the Italian context.

The model works for individual age cohorts of the population selected for vaccination, using a Markovian structure. The population can also be disaggregated by risk level.

The Figure 1 summarises the states considered in the Markov model.

In adapting the model to the Italian context, following the current PNPV guidelines, the effects of vaccination with PCV20 were compared with the vaccination strategy envisaging the administration of 13-valent pneumococcal conjugate vaccine (PCV13). For completing the analysis, comparisons were also made with 15-valent pneumococcal conjugate vaccine (PCV15). The PCV15 was recently approved by the European Commission (EC) for active immunization for the prevention of invasive disease and pneumonia caused by *Streptococcus pneumoniae* in individuals 18 years of age and older.

As far as age groups are concerned, the analyses were conducted envisaging vaccination of the cohorts of subjects 65 to 74 years of age.

Some information is also provided on the impact of vaccination of the 65 years of age cohort alone. These estimates provide an indication of the economic and financial burden actually incurred by the NHS.

A lifetime horizon was adopted in the simulations; costs and (health) consequences were discounted at a 3% discount rate per year.

As far as costs are concerned, in adapting the model to the Italian context, direct health costs were taken into account from the NHS perspective.

The efficiency evaluations of the model were expressed both in cost per life year gained (cost-effectiveness) and per QALY gained (cost utility). 

### 2.1. Population

The model is populated with the adult population (yearly average) inferred from Italian National Institute of Statistics (Istat) statistics. In the absence of Italian evidence regarding the Italian distribution by risk levels, the composition (by age group) surveyed by the Center for Disease Control and Prevention (CDC) in the USA [14] was used. Risk levels are defined as follows [15]:Low risk: immunocompetent people without chronic clinical conditions;Medium risk: immunocompetent people with at least 1 chronic clinical condition (such as cardiovascular, liver, pulmonary disease, diabetes, asthma, smoking, and alcohol abuse);High risk: immunocompromised people due to chronic renal failure or neoplasm (including leukaemia, lymphoma, and solid tumours diagnosed less than 3 years ago).

Due to the absence of information on transition probabilities among risk level for the Italian population, in the simulations it was assumed that there is no switching between risk levels.

### 2.2. Past Vaccination Coverage

For the sake of simplicity, the model assumes that none of the adults in the cohort considered have previously received PCV13: the hypothesis is also supported by the relatively short time elapsed since the start of vaccinations in children (7-valent pneumococcal conjugate vaccine was approved in 2005 and then replaced by PCV13 in 2010). 

### 2.3. Incidence of Disease

The model distinguishes between invasive pneumococcal disease (IPD), disaggregating meningitis from the other forms of bacteraemia, and the other types of nonbacteremic pneumonia (NBP), disaggregated into hospitalised and non-hospitalised ones. In the model, disease incidences are used for determining transition to the above-mentioned states (Table 1).

As to IPD, reference was made to the surveys of the Istituto Superiore di Sanità (ISS) [3]; the 2019 survey was used since the 2020 data were influenced by the pandemic emergency.

With regard to NBP, reference was made to the evidence inferable from the Ministry of Health [16] surveys on hospital activity (flow of hospital discharge forms (SDO) year 2019).

In particular, all hospitalisations (ordinary ones and outpatients in day hospital settings) with ICD-IX diagnoses from 480 to 486 were extracted, detecting an overall incidence of 751.0 ordinary cases per 100,000 inhabitants aged over 64 years. The incidences detected (also by age group) were then corrected by risk level on the basis of the information provided, for the years 2017–2018, by the CDC [17]. Literature evidence [18,19,20] was used to estimate the share of non-hospitalised NBP cases.

### 2.4. Composition of Disease Cases by Serotype

With regard to the attribution of disease cases to the different serotypes, for invasive pneumococcal diseases (IPDs) reference was made to the data collected (year 2019, 2020) by the ISS [3]; for pneumococcal CAP (P-NBP) evidence from the PUMA study [21] was used, considering a proportion of P-NBP of 15.1% for the population aged 65–74 years (Table 2).

### 2.5. Mortality

As for the general population mortality, reference was made to the biometric tables published by Istat.

Mortality for IPD (separately for meningitis and other types of bacteraemia) was inferred from literature [22].

Mortality related to hospitalised NBP cases (by age) was processed using the hospital discharge forms (SDO) published by the Ministry of Health (for the year 2019) [16]. Mortality for non-hospitalised cases (that in Italy are largely underestimated) reasonably is very low; unfortunately no evidence was founded for Italy, consequently it was prudentially assumed to be one-tenth of the mortality rate for the corresponding hospitalised cases.

In the absence of evidence on mortality by risk level for the Italian population, it was prudentially assumed to be constant (Table 3).

### 2.6. Quality of Life (Utility)

With specific reference to the quality of life (QoL), reference was made to the work of Sisk et al. [23] as cited by Boccalini et al. [22]: QoL comes from an elaboration on the USA population Activities of Daily Living (ADL) resulting from the National Health Interview Survey (1990), applying a multi-attribute utility scaling technique.

Disutilities applied to cases of IPD and NBP were derived from the work of Boccalini et al. [22] that analysed the economic impact of pneumococcal vaccination strategies specifically in the Italian adults (Table 4). Their approach may underestimate the QALYs gap since it considers only the disutilities generated during the hospitalisation period. Other studies adopted a more extensive approach: for example, Mangen et al. [24] quantified in 0.13 the QALY difference (on an annual basis) in two cohorts, diseased and non diseased subjects, based on hospitalisation with suspected community-acquired pneumonia.

### 2.7. Healthcare Costs

The prices of the vaccines adopted are equal to the present (EUR 2022) maximum cost of transfer to the NHS (i.e., the price paid by the Italian public NHS to manufacturer).

The cost of administration can vary at regional level, due to the federal organization of the Italian NHS the published tariff by the Emilia Romagna Region [25] was used, as the region is a reference at the national level (Table 5).

Regarding hospitalisation costs (Table 6), the average fees processed (by age class) on the basis of the Diagnosis Related Groups (DRGs) associated with ordinary hospitalisations (SDO, Ministry of Health) were assumed; for non-hospitalised NBP cases, in absence of an estimate of the consultation in primary care (Italian general practitioners are paid on a capitation basis), a value equal to the value of the fees applied to the DRGs associated with outpatients in day hospital settings was assumed.

### 2.8. Vaccine Efficacy

With specific reference to the efficacy of vaccines against IPD and NBP (Table 7), for the low/medium risk population, the reference used is the evidence produced by the Community-Acquired Pneumonia Immunization Trial in Adults (CAPiTA) [26,27].

For the high-risk population, vaccine efficacy was assumed to be 80% of the corresponding values for the low/medium risk population based on evidence from the work of Klugman et al. [28].

The efficacy of conjugate vaccines is assumed to be constant for the first 5 years and is then reduced by 5% per year for the following 5 years (up to year 10), by 10% between years 11 and 15. It is finally nullified after year 16.

### 2.9. Sensitivity Analysis

A one-way deterministic and a probabilistic sensitivity analysis was performed.

Simulations were elaborated on the principal variables affecting results; specifically: Incidence of the disease;Disutilities;Mortality rate;Effectiveness of PCV;Medical costs.

For the one-way sensitivity analysis a variation of ±25% was assumed, except disease disutilities, assumed to vary in the range between 0.01 (Boccalini et al. [22]) and 0.13 (Mangen et al. [24]).

For the probabilistic analysis a Beta distribution was assumed for disease incidence parameter, mortality, and effectiveness of vaccines, while a Gamma distribution was used for medical costs. 

## 3. Results

A total of 13.5% of the Italian population, i.e., 6,795,374, are aged between 65 and 74 years, of whom 48% are assumed to have a low risk, 40.6% a medium risk, and the remaining 11.4% a high risk (see definition and caveat expressed in paragraph 2.1).

Assuming complete (100%) vaccination of cohorts aged 65–74 years, in the (lifetime) comparison between PCV20 and PCV13, the former is dominant (lower incremental cost for a better health outcome).

A reduction in disease events is estimated: −1208 deaths; −1171 cases of bacteraemia (excluding meningitis); −227 cases of meningitis; −9845 hospitalised NBP cases; and −21,058 non-hospitalised NBP cases.

Overall, in the Italian population, a total benefit gain of 6581.6 in terms of life years and of 4734.0 in terms of QALY is estimated.

On the cost side, against an increase in costs due to vaccinations equal to EUR 40.568 million, other direct health costs are reduced by EUR 48.032 million, with a net savings of EUR +7.464 million (Table 8).

In the case of vaccination of the 65 years of age cohort alone, as envisaged by the PNPV, the cost-effectiveness of PCV20 vs. PCV13 is confirmed, with an Incremental Cost-Effectiveness Ratio (ICER) of EUR 196 per life year gained and EUR 268 per QALY gained.

On the cost side, the increase in costs due to vaccinations is EUR 4.946 million and the other direct health costs are reduced by EUR 4.817 million, with an incremental net cost of EUR +0.129 million.

Assuming vaccination of the cohorts aged 65–74 years, the comparison between PCV20 and PCV15 results in an ICER of EUR 66 per life year gained and EUR 91 per QALY gained.

The estimated reduction in disease events is −1009 deaths; −940 cases of bacteraemia (excluding meningitis); −183 meningitis; −8350 hospitalised NBP cases; −17,858 non-hospitalised NBP cases, with an overall gain of 5536.7 life years and 3984.7 QALYs.

On the cost side, against an increase in costs due to vaccinations equal to EUR 40.568 million, the other direct health costs decreased by EUR 40.205 million, with an incremental net cost of EUR +0.364 million.

### Sensitivity Analysis

Table 9 reports the Incremental cost per QALY gained resulting from the one-way sensitivity analysis.

The simulations demonstrate that results are quite robust: PCV20 remains dominant vs. PCV13 and PCV15 in most of the simulations, and in the remaining is cost-effective assuming a very low willingness to pay.

The probabilistic sensitivity analysis (Figure 2) confirms the results’ robustness: although in Italy no explicit thresholds are available, the probability of PCV20 to be cost-effective also at a very modest threshold of EUR 5000 exceeds 90% in all the simulations proposed.

## 4. Discussion

In 2020, EUR 106.5 million were spent for pneumococcal vaccination (children and adults), of which 93.5% was for PCV13, 2.4% was for PCV10 (used for children), and 4.2% was for 23-valent pneumococcal polysaccharide vaccine (PPV23) [29].

Assuming a cost for PCV20 of EUR 55.97 per dose to be borne by the NHS, the changeover of vaccinations from PCV13 to PCV20, for the 65–74 years of age cohorts (not considering the other subjects at risk), implies a financial burden for the NHS (in other terms an increase in NHS costs due to the immunization with PCV20 instead of PCV13) equal to EUR 40.568 million (including administration costs, which are an ongoing expense and would be consistent regardless of which PCV was administered).

Although the other direct health costs are reduced by about EUR 48.032 million, particularly due to the reduction in hospitalisations due to IPD and NBP, thus providing a net savings of EUR 7.464 million in the long term, this is a strategy that is difficult to implement in the NHS [30]; in fact, the financial burden it would generate in the short term is expected to be not sustainable: nonetheless, our results assess the significant dimension of the improvement in health outcomes that can be potentially obtained with an increase in the investment in immunisation.

Against the increase in burdens on the NHS, the health effects are indeed appreciable: deaths attributable to *S. pneumoniae* are reduced by 1208 (−8.7 per 100,000 inhabitants aged over 64 years); the cases of bacteraemia, excluding meningitis are reduced by 1171 (−8.4 per 100,000 inhabitants aged over 64 years). The cases of meningitis are reduced by 227 (−1.6 per 100,000 inhabitants aged over 64 years); hospitalised NBP cases are reduced by 9845 (−71 per 100,000 inhabitants aged over 64 years) and non-hospitalised NBP cases are reduced by 21,058 (−151.9 per 100,000 inhabitants aged over 64 years).

The reduction in deaths results in an overall gain in life years equal to 6581.6, and a gain in quality of life due to the reduction in hospitalisation-related disutilities, equal to 4734 QALYs. The results are likely to be conservative due to the assumptions made in relation to the disease disutilities in the base case.

Overall, PCV20 is dominant over PCV13, having lower costs and better health outcomes, thus demonstrating the social efficiency of the investment. Results are very robust as indicated by one-way and probabilistic sensitivity analysis.

Limiting immunisation to the 65 years of age cohort only, as envisaged by the PNPV (but not considering the other subjects at risk), the switch from PCV13 to PCV20 implies a cost for the National Health Service equal to EUR 4.946 million (i.e., 4.6% of the current spending on pneumococcal vaccines). The other direct health costs are reduced by EUR 4.817 million, particularly due to the decrease in IPD and NBP hospitalisations, thus bringing the net burden to EUR +0.129 million: an amount that is largely sustainable, being equal to 1.2 per thousand of the current spending on pneumococcal vaccines.

We can therefore conclude that the switch from PCV13 to PCV20 vaccination is certainly a sustainable investment, as well as efficient for the NHS, assuming modest willingness-to-pay thresholds compared with those normally used at regulatory level in Italy.

Moreover, with respect to the use of PCV15, assuming for it a price equal to that of PCV13, the results obtained are confirmed: the ICER of PCV20 compared with PCV15 is EUR 66 per life year gained and EUR 91 per QALY gained. PCV20 is therefore cost-effective even assuming a very low social willingness to pay. The financial differential of burden on the NHS remains unchanged with respect to PCV13, since the same price was assumed for PCV15 as for PCV13.

The importance of pneumococcal vaccination in adults has been the subject of numerous analyses, and is now an established fact, as can be seen from the Italian PNPV, which recommends it for adults aged 65 years, as well as for individuals at risk. The current PNPV recommends the use of PCV13, followed by vaccination with the PPV23 after at least two months.

Unfortunately, very limited evidence is available on the cost-effectiveness of the new PCV20. The present paper, to our knowledge, is one of the very first attempt to produce an assessment of the economic impact derived by a strategy of substitution of PCV13 immunization with PCV20.

The model suffers from a number of limitations linked, in particular, to the lack of data for the Italian population: first and foremost, the limitation of adopting a risk composition of the population deduced from US data, as well as levels of utility (and disutility) inferred from literature, not specific to the Italian population.

The production of the cited figures should be a priority in terms of public health planning: in the meanwhile, the strategy used in the paper for “missing” data was to adopt a conservative approach.

Moreover, in the absence of disaggregated data on vaccination in adults and children, it appears evident that in Italian practice, the use of PPV23 after PCV13 in adults is substantially a marginal choice. Given its lack of use in actual Italian practice, PPV23 was not considered in the analysis.

Finally, the financial impact of the assessments made is underestimated due to the fact that the needs of the subjects at risk are not taken into account. However, the clinical effects are also proportionally underestimated so that it can be assumed that the results obtained in terms of ICER remain valid even in the real context of the implementation of the vaccination strategy.

## 5. Conclusions

The Italian National Vaccine Prevention Plan recommends pneumococcal vaccination for adults aged 65 years, as well as for individuals at risk.

Although the coverage achieved in adults is not known, it is assumed that it is not yet optimal.

The availability of the 20-valent conjugate vaccine provides an opportunity to prevent additional pneumococcal disease. The cost-effectiveness (cost utility) model, adapted to the Italian context used to assess the switch to PCV20 (from PCV13 and/or PCV15) confirms the significance of the achievable health benefits. The vaccination strategy according to the PNPV (vaccination of the cohort aged 65 years) is not only efficient (cost-effective) by adopting a low threshold of (social) willingness to pay, but is also largely sustainable, with a limited impact on the current national budget for vaccines.

## Figures and Tables

**Figure 1 vaccines-10-02032-f001:**
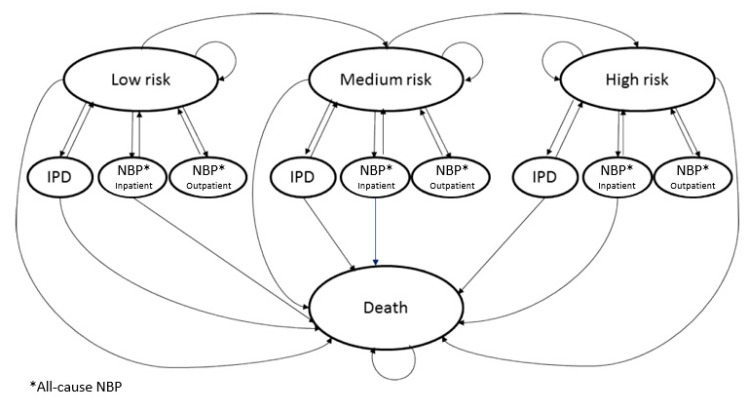
Model structure. Note: IPD: invasive pneumococcal disease; NBP: all-cause nonbacteremic pneumonia.

**Figure 2 vaccines-10-02032-f002:**
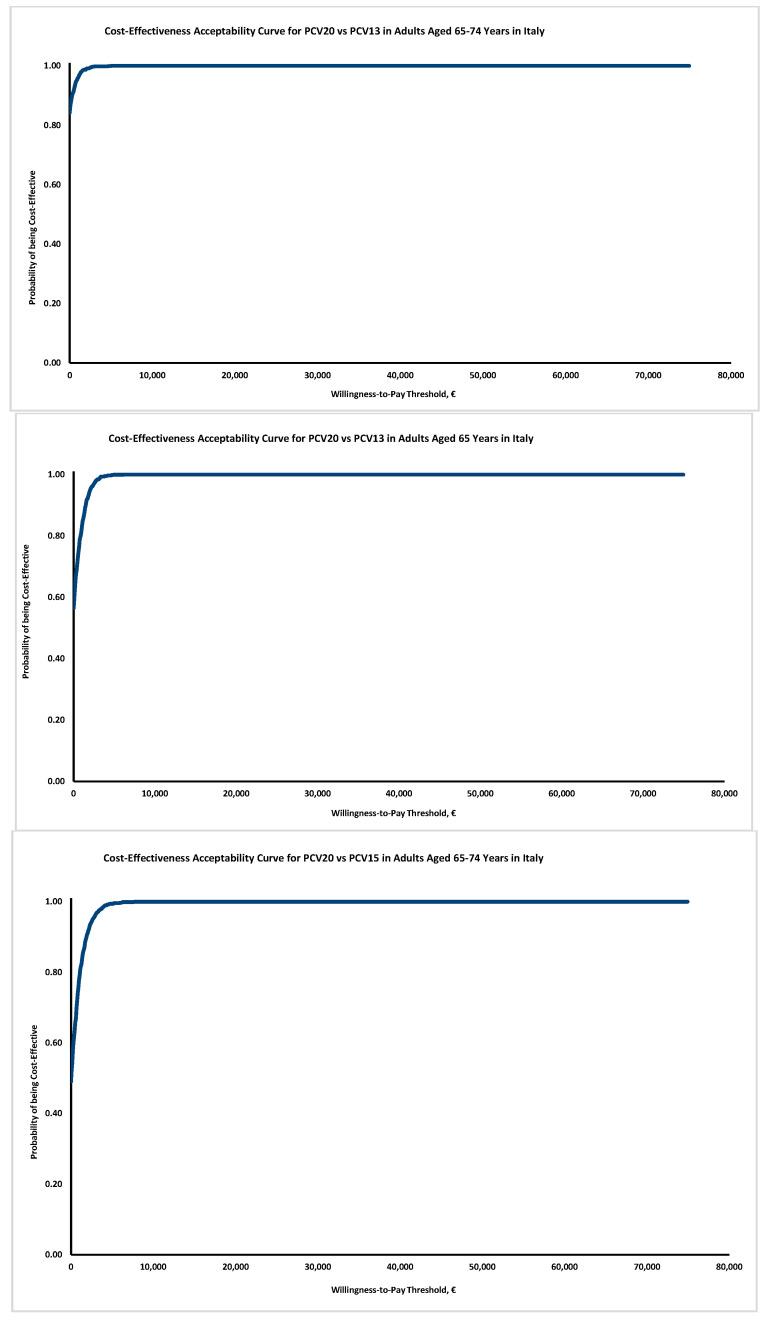
Cost-effectiveness acceptability curve.

**Table 1 vaccines-10-02032-t001:** Disease incidence (people aged >64 years): rates per 100,000 inhabitants.

Bacteraemia (Excluding Meningitis)	Meningitis	Hospitalised NBP Cases	Non-Hospitalised NBP Cases
6.134 [3]	1.183 [3]	751.006 [16]	1610.678 [18,19,20]

**Table 2 vaccines-10-02032-t002:** Share of cases by serotype.

Disease	PCV13	PCV15	PCV20
IPD [3]	31.6%	39.1%	69.6%
P-NBP [21]	25.0%	33.6%	57.2%

**Table 3 vaccines-10-02032-t003:** Mortality (% values).

Age	General Population (All Cause)	Bacteraemia excluding Meningitis	Meningitis	Hospitalised NBP Cases	Non-Hospitalised NBP Cases
65–74	1.489	9.200 [22]	18.500 [22]	5.386 [16]	0.539

**Table 4 vaccines-10-02032-t004:** Quality of life (HRQoL).

Age	Utility	Weighted Annual Disutility			
	General Population	Bacteraemia Excluding Meningitis	Meningitis	Hospitalised NBP Cases	Non-Hospitalised NBP Cases
65–74	0.75 [23]	0.02 [22]	0.02 [22]	0.01 [22]	0.01 [22]

**Table 5 vaccines-10-02032-t005:** Cost of vaccines *.

Vaccine	Cost of Dose (EUR)	Admin.tion. (EUR) [25]
PCV13	50.00	16.00
PCV15	50.00	16.00
PCV20	55.97	16.00

* Maximum cost of transfer to the NHS.

**Table 6 vaccines-10-02032-t006:** Other healthcare costs (EUR).

Age	Hospitalisations			Costs Assoc.ed with Outpatients
	Bacteraemia Excluding Meningitis	Meningitis	NBP	NBP
65–74	12,192	12,192	3764	419

**Table 7 vaccines-10-02032-t007:** Initial vaccine efficacy 65–74 age group (% values).

Risk	IPD	NBP
Low [26,27]	75.0	45.0
Medium [26,27]	75.0	45.0
High [28]	60.0	36.0

**Table 8 vaccines-10-02032-t008:** Main results.

Vaccine	PCV13	PCV15	PCV20	PCV20 vs. PCV13	PCV20 vs. PCV15
Disease cases					
Bacteraemia (excluding meningitis)	4982	4751	3811	−1171	−940
Meningitis	965	921	738	−227	−183
Hospitalised NBP cases	771,594	770,098	761,748	−9845	−8350
Non-hospitalised NBP cases	1,651,431	1,648,231	1,630,373	−21,058	−17,858
Outcomes					
Deaths (from IPD and NBP)	99,715	99,516	98,506	−1208	−1009
Life years (LY) (×1000)	75,044.2	75,042.2	75,050.7	+6.6	+5.5
QALY (×1.000)	52,687.2	52,688.0	52,692.0	+4.7	+4.0
Costs					
Vaccination costs (× EUR 1000)	EUR 448,495	EUR 448,495	EUR 489,063	EUR +40,568	EUR +40,568
Other health costs (× EUR 1000)	EUR 2,523,553	EUR 2,515,726	EUR 2,475,521	EUR −48,032	EUR −40,205
Total health costs (× EUR 1000)	EUR 2,972,047	EUR 2,964,220	EUR 2,964,584	EUR −7464	EUR −0.364
ICER					
ICER for LY gained (EUR)				Dominant	66
ICER for QALY gained (EUR)				Dominant	91

**Table 9 vaccines-10-02032-t009:** One-way sensitivity analyses for cost-effectiveness.

Parameter	Incremental Cost per QALY Gained					
	PCV20 vs. PCV13 Adults Aged 65–74 Years (N = 6,795,374)		PCV20 vs. PCV13 Adults Aged 65 Years (N = 828,558)		PCV20 vs. PCV15 Adults Aged 65–74 Years (N = 6,795,374)	
	Lower Bound *	Upper Bound *	Lower Bound *	Upper Bound *	Lower Bound *	Upper Bound *
Disease Incidence						
Bacteraemia	Dominant	Dominant	EUR 576	Dominant	EUR 597	Dominant
Meningitis	Dominant	Dominant	EUR 31	Dominant	EUR 188	Dominant
Inpatient All-Cause NBP	Dominant	Dominant	EUR 1489	Dominant	EUR 1954	Dominant
Outpatient All-Cause NBP	Dominant	Dominant	EUR 235	Dominant	EUR 487	Dominant
Disutility						
Disutility due to Bacteraemia	Dominant	Dominant	Dominant	Dominant	EUR 92	EUR 89
Disutility due to Meningitis	Dominant	Dominant	Dominant	Dominant	EUR 91	EUR 91
Disutility due to Inpatient All-Cause NBP	Dominant	Dominant	Dominant	Dominant	EUR 93	EUR 76
Mortality						
Bacteraemia Case Fatality	Dominant	Dominant	Dominant	Dominant	EUR 92	EUR 91
Meningitis Case Fatality	Dominant	Dominant	Dominant	Dominant	EUR 91	EUR 91
Inpatient All-Cause NBP Case Fatality	Dominant	Dominant	Dominant	Dominant	EUR 88	EUR 94
Outpatient All-Cause NBP Case Fatality	Dominant	Dominant	Dominant	Dominant	EUR 91	EUR 92
Vaccine Effectiveness						
Bacteraemia/Meningitis	Dominant	Dominant	EUR 1405	Dominant	EUR 704	Dominant
Inpatient/Outpatient All-Cause NBP	EUR 439	Dominant	EUR 4370	Dominant	EUR 2564	Dominant
Medical Costs						
Bacteraemia	Dominant	Dominant	EUR 556	Dominant	EUR 581	Dominant
Meningitis	Dominant	Dominant	EUR 31	Dominant	EUR 187	Dominant
Inpatient All-Cause NBP	Dominant	Dominant	EUR 1272	Dominant	EUR 1650	Dominant
Outpatient All-Cause NBP	Dominant	Dominant	EUR 227	Dominant	EUR 469	Dominant

* Upper and lower bound equal to ±25%, except disutilities for which the range 0.01 (Boccalini et al. [20]), 0.13 (Mangen et al. [22]) was used.

## Data Availability

The model presented in this study are available on reasonable request from the corresponding author.

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
