# Peer review of "Cost-Effectiveness of Vaccination with the 20-Valent Pneumococcal Conjugate Vaccine in the Italian Adult Population"

_vaccines, 2022, doi:10.3390/vaccines10122032_

Round 1
Reviewer 1 Report
Abstract: first use of NBV, please provide the full name.
Introduction
References are required for the WHO and CDC evidence in the second paragraph. In fact, more references are required throughout the Introduction, for example:
- It is estimated that 20 to 40% of children and 5 to 10% of adults are asymptomatic carriers of the bacterium in the nasopharynx, the only known reservoir in humans
- Most pneumococcal infections are mild; some of them, however, can be fatal or cause long-term problems.
I do not understand why the authors provide Italian incidence numbers and rates for 2020. 2019 provides a reliable estimate that is more applicable to other years than 2020.
Pg 2 line 3: please define IPD.
For Figure 1, please provide a footnote with all of the abbreviations spelled out in full.
Pg 3, 2.1 Population: please provide a reference for Istat.
Can the authors please explain why the cohort cannot move between risk levels. As it is an older cohort, it seems logical that people will move from low to medium risk, and on to high risk.
I cannot see any mention of transition probabilities. Please include a table with these.
Pg 5 states: Mortality for non-hospitalised cases is considered very low in literature: it has therefore been assumed to be one-tenth of the mortality rate for the corresponding hospitalised cases. If there is literature citing mortality rates, why is an assumption on 1/10 applied?
Pg 5 states: In the absence of evidence on mortality by risk level, it was assumed to be constant. I assume that mortality rates would be substantially higher for the high-risk group than the low and medium risk groups. Can this be explored in sensitivity analyses?
Table 3 requires clarification. The heading is “Mortality (% values)”. Does this mean that 1.479% of the general population aged 65-74 will die from a pneumococcus infection? I’m unclear regarding the denominator that is used.
Regarding quality of life, the authors have based this on the work of Sisk. Sisk, in turn referenced work from Torrance and Erickson. Can the authors please provide more information about the utilities used, the instruments, population from which they were derived.
The disutilites in Table 4 are surprisingly small: 0.02 for meningitis. For example, see O’Reilly et al (2022) The impact of acute pneumococcal disease on health state utility values: a systematic review. Qual Life Research.
In general, please provide references of source data in the tables.
2.7 Health care costs states: The prices of the vaccines adopted are equal to the maximum cost of transfer to the NHS. The cost of administration has been assumed to be equal to the fee published by the Emilia Romagna Region. Please clarify what this means – I’m not familiar with the Italian health system.
Do the outpatient costs include consultation in primary care?
Please state the currency and year.
Were tornado diagrams generated to inform the sensitivity analyses?
What does mortality/fatality refer to? Should this just be the mortality rate?
Results:
Pg 6 states: …..of whom 48% are assumed to have a low risk, 40.6% a medium risk and the remaining 11.4% a high risk. Please clarify what the risk is.
Please clarify that these are ICERs: Overall, in the Italian population, a total gain of 6,581.6 in terms of life years and of 4,734.0 in terms of QALY is estimated.
I see that vaccine coverage is 100%. Can the authors please comment of how the ICER was impacted when decreasing this in-line with published data (either for this vaccine or for the age group).
Table 8: please add Euros to the column headings. Please review the headings: Vaccine should be above the vaccines, and Outcomes above the left hand column. I suggest the authors look at other ways of reporting these results, in-line with accepted approaches. This is not a reader-friendly format.
Please define the willingness to pay threshold: it is mentioned in the sensitivity analyses but nor figure is provided.
Discussion
Overall, this section could be structured to provide the reader with a clear overview of the main results, how they compare to other studies, policy implications and limitations.
The third paragraph …. Even in the absence of disaggregated data…. This belongs in the limitations or further down in this section. I suggest telling the reader your main results in the first paragraph.
Paragraph 4 states: implies a financial burden for the NHS equal to 40.568 million euros…. Is this cost in addition to what the NHS is/will pay for PVC13?
This is a long sentence (below). Please consider revising and discussing this in more detail. For example, PVC20 is cost-effective over the longer term, but upfront cost means that it may not be acceptable to the NHS.
Although the other direct health costs are reduced by about 48.032 million euros, particularly due to the reduction in hospitalisations due to IPD and NBP, thus providing a net savings of 7.464 million euros, this is a strategy that is difficult to implement in the NHS [28], due to the financial burden it would generate: nonetheless, it appears to be a useful benchmark to assess the improvement in health outcomes that can be obtained with an investment in immunisation.
The limitations section can be explored in more detail: what impacts do the authors thinks that these factors may have on the results?
Author Response
First of all, all the authors would like to thank the referee for the punctual observations that contribute significantly to the improvement of the quality of the work.
Below we give feedback point by point to the observations received, reporting the changes to the manuscript made: we hope to have properly answered to all the observations.
Thanks again
Referee observations
Abstract: first use of NBV, please provide the full name. DONE
Introduction
References are required for the WHO and CDC evidence in the second paragraph. DONE
In fact, more references are required throughout the Introduction, for example:
- It is estimated that 20 to 40% of children and 5 to 10% of adults are asymptomatic carriers of the bacterium in the nasopharynx, the only known reservoir in humans INSERTED
- Most pneumococcal infections are mild; some of them, however, can be fatal or cause long-term problems. INSERTED
I do not understand why the authors provide Italian incidence numbers and rates for 2020. 2019 provides a reliable estimate that is more applicable to other years than 2020. Clarified that numbers and rate are produced for both the two years, and 2020 is biased by the pandemia.
Pg 2 line 3: please define IPD. DONE
For Figure 1, please provide a footnote with all of the abbreviations spelled out in full. DONE
Pg 3, 2.1 Population: please provide a reference for Istat. DONE
Can the authors please explain why the cohort cannot move between risk levels. As it is an older cohort, it seems logical that people will move from low to medium risk, and on to high risk. Unfortunately, we were not able to find any evidence about the risk stratification of the Italian population, including transition probabilities: consequently, we adopted a conservative approach avoiding switching
I cannot see any mention of transition probabilities. Please include a table with these. We have specified that the other transition probabilities are referred in the following sections (mainly incidence of disease and mortality)
Pg 5 states: Mortality for non-hospitalised cases is considered very low in literature: it has therefore been assumed to be one-tenth of the mortality rate for the corresponding hospitalised cases. If there is literature citing mortality rates, why is an assumption on 1/10 applied? Non-hospitalized cases are largely underestimates and we only found qualitative assumption presuming that mortality for these cases are not significantly different from general population. We only have an Italian data for hospitalized cases, and so we decided to apply a prudential estimate.
Pg 5 states: In the absence of evidence on mortality by risk level, it was assumed to be constant. I assume that mortality rates would be substantially higher for the high-risk group than the low and medium risk groups. Can this be explored in sensitivity analyses? Unfortunately, the model is very complex and simulation requires very heavy calculation: at present simulation for risk group are not routinely allowed.
Table 3 requires clarification. The heading is “Mortality (% values)”. Does this mean that 1.479% of the general population aged 65-74 will die from a pneumococcus infection? I’m unclear regarding the denominator that is used. Clarified that the figure refers to all cause mortality (death not due to pneumococcal disease) for the Italian general population aged 65-74; other columns refer to mortality due to illness.
Regarding quality of life, the authors have based this on the work of Sisk. Sisk, in turn referenced work from Torrance and Erickson. Can the authors please provide more information about the utilities used, the instruments, population from which they were derived. We used the quality of life already used by Boccalini et al. (2013), as is the only reference for QoL already applied to the Italian population; we added a clarification in the text regarding the reference. Sisk quotes Torrance and Erickson… and they produced data for the USA population.
The disutilites in Table 4 are surprisingly small: 0.02 for meningitis. For example, see O’Reilly et al (2022) The impact of acute pneumococcal disease on health state utility values: a systematic review. Qual Life Research. That is the calculation provided by Boccalini et al. (2013): they start from a spot disutility of 0.2 for people with the disease, applying an average of 34 days of hospitalization following Italian clinical practice… consequently, the “weighted annual” disutility result in the cited 0.02. In the sensitivity analysis we used Mangen (0.13) to test the impact of the assumption.
In general, please provide references of source data in the tables. DONE
2.7 Health care costs states: The prices of the vaccines adopted are equal to the maximum cost of transfer to the NHS. The cost of administration has been assumed to be equal to the fee published by the Emilia Romagna Region. Please clarify what this means – I’m not familiar with the Italian health system. We have added some more details: hoping to have clarified
Do the outpatient costs include consultation in primary care? Clarified in the manuscript
Please state the currency and year. DONE
Were tornado diagrams generated to inform the sensitivity analyses? Tornado diagrams are not very meaningful in the specific case, as most of the sensitivity analysis confirmed the dominance of PCV20: we think the table is more appropriate
What does mortality/fatality refer to? Should this just be the mortality rate? Yes corrected
Results:
Pg 6 states: …..of whom 48% are assumed to have a low risk, 40.6% a medium risk and the remaining 11.4% a high risk. Please clarify what the risk is. Inserted a refer to the Matherials & Methods, Population paragraph
Please clarify that these are ICERs: Overall, in the Italian population, a total gain of 6,581.6 in terms of life years and of 4,734.0 in terms of QALY is estimated. Reformulated to clarify the “benefit” nature of the cited results, and following of the cost and ICER estimates.
I see that vaccine coverage is 100%. Can the authors please comment of how the ICER was impacted when decreasing this in-line with published data (either for this vaccine or for the age group). While changing the uptake rate will change the total QALYs gained and total costs, we do not think it changes the ICER in the model unless uptake is varied across risk groups. Moreover, from a health policies point of view, consider that Italian immunization plan provides indication to vaccinate all people of the 65 cohort, as well as all the people at “high risk” and, as commented in the manuscript, immunization of the 65-74 cohort is largely considered financially unsustainable; but we’ve decided to simulate both the alternatives, because the 65-74 cohort seems to be a good quantitative proxy of the population potentially eligible to gain benefit from the immunization, and comparing the two permits to appreciate the “costs” consequent to the financial constraint.
Table 8: please add Euros to the column headings. Please review the headings: Vaccine should be above the vaccines, and Outcomes above the left hand column. I suggest the authors look at other ways of reporting these results, in-line with accepted approaches. This is not a reader-friendly format. DONE
Please define the willingness to pay threshold: it is mentioned in the sensitivity analyses but nor figure is provided. DONE
Discussion
Overall, this section could be structured to provide the reader with a clear overview of the main results, how they compare to other studies, policy implications and limitations. We hope to have improved the section
The third paragraph …. Even in the absence of disaggregated data…. This belongs in the limitations or further down in this section. I suggest telling the reader your main results in the first paragraph. OK thanks
Paragraph 4 states: implies a financial burden for the NHS equal to 40.568 million euros…. Is this cost in addition to what the NHS is/will pay for PVC13? Clarified
This is a long sentence (below). Please consider revising and discussing this in more detail. For example, PVC20 is cost-effective over the longer term, but upfront cost means that it may not be acceptable to the NHS. Revised, thanks
Although the other direct health costs are reduced by about 48.032 million euros, particularly due to the reduction in hospitalisations due to IPD and NBP, thus providing a net savings of 7.464 million euros, this is a strategy that is difficult to implement in the NHS [28], due to the financial burden it would generate: nonetheless, it appears to be a useful benchmark to assess the improvement in health outcomes that can be obtained with an investment in immunisation.
The limitations section can be explored in more detail: what impacts do the authors thinks that these factors may have on the results? We hope to have properly integrated
Reviewer 2 Report
This study aimed to assess the cost-effectiveness of PCV20 vaccination compared to PCV13 and PCV15 administration in Italy. This research is important, especially because vaccination is currently a common program implemented by almost all countries in the world due to the COVID-19 pandemic. The strength of this study is related to the model framework and quantitative modeling used in assessing the cost-effectiveness of vaccination.
Comments
In the introduction section, the author has explained the cases of infection worldwide; however, it has not disclosed the implementation and vaccination policies in various countries worldwide. In this context, the authors are also advised to disclose previous studies on the cost-effectiveness of vaccination. The author must also identify knowledge gaps related to this research topic in the introduction section.
The author needs to add a separate section in the material and methods section to explain the operational definitions of variables and the terms and abbreviations used to make it easier for readers.
The author has not compared his findings with those of similar studies in the discussion section. It is hoped that the authors will further explore the results of similar studies, especially related to the methods used and the cost-effectiveness of vaccination.
The conclusions provided are appropriate and consistent with the evidence and arguments presented. However, the authors have not disclosed theoretical recommendations as well as proposals for future research.
The references used do not refer to the latest references (the last five years). Authors need to add the latest references, especially from articles sourced from reputable journals.
Author Response
First of all, all the authors would like to thank the referee for the indication that contributes significantly to the improvement of the quality of the work.
Below we give feedback point by point to the observations received, reporting the changes to the manuscript made: we hope to have properly answered to all the observations.
Thanks again
Referee observations
This study aimed to assess the cost-effectiveness of PCV20 vaccination compared to PCV13 and PCV15 administration in Italy. This research is important, especially because vaccination is currently a common program implemented by almost all countries in the world due to the COVID-19 pandemic. The strength of this study is related to the model framework and quantitative modeling used in assessing the costeffectiveness of vaccination.
Comments
In the introduction section, the author has explained the cases of infection worldwide; however, it has not disclosed the implementation and vaccination policies in various countries worldwide. In this context, the authors are also advised to disclose previous studies on the cost-effectiveness of vaccination. The author must also identify knowledge gaps related to this research topic in the introduction section. DONE
The author needs to add a separate section in the material and methods section to explain the operational definitions of variables and the terms and abbreviations used to make it easier for readers. We double checked abbreviations completing the explanation and inserted the variables description in Matherial & Methods
The author has not compared his findings with those of similar studies in the discussion section. It is hoped that the authors will further explore the results of similar studies, especially related to the methods used and the cost-effectiveness of vaccination. Some more details have been included: please consider that, to our knowledge is one of the very first attempt to produce quantitative assessment of the PCV20 vaccination efficiency
The conclusions provided are appropriate and consistent with the evidence and arguments presented. However, the authors have not disclosed theoretical recommendations as well as proposals for future research. Hoping to have improved the manuscript stressing the consideration on the efficiency of the adoption of PCV20, both in presence (see the 65 only cohort simulation, aligned with the Italian National Immunization Plan) and absence of financial constraint.
The references used do not refer to the latest references (the last five years). Authors need to add the latest references, especially from articles sourced from reputable journals We quoted a dozen of reference after 2016: please consider that is one of the very first attempt to produce cost-effectiveness assessment on the PCV20 vaccination